# Structural mechanism of proton conduction in otopetrin proton channel

Ninghai Gan [1,2], Weizhong Zeng[1,2], Yan Han [2], Qingfeng Chen[3] & Youxing Jiang [1,2] ✉

The otopetrin (OTOP) proteins were recently characterized as extracellular proton-activated proton channels. Several recent OTOP channel structures demonstrated that the channels form a dimer with each subunit adopting a double-barrel architecture. However, the structural mechanisms underlying some basic functional properties of the OTOP channels remain unresolved, including extracellular pH activation, proton conducting pathway, and rapid desensitization. In this study, we performed structural and functional characterization of the *Caenorhabditis elegans* OTOP8 (CeOTOP8) and mouse OTOP2 (mOTOP2) and illuminated a set of conformational changes related to the proton-conducting process in OTOP. The structures of CeOTOP8 reveal the conformational change at the N-terminal part of TM12 that renders the channel in a transiently proton-transferring state, elucidating an inter-barrel, Glu/His-bridged proton passage within each subunit. The structures of mOTOP2 reveal the conformational change at the N-terminal part of TM6 that exposes the central glutamate to the extracellular solution for protonation. In addition, the structural comparison between CeOTOP8 and mOTOP2, along with the structure-based mutagenesis, demonstrates that an inter-subunit movement at the OTOP channel dimer interface plays a central role in regulating channel activity. Combining the structural information from both channels, we propose a working model describing the multi-step conformational changes during the proton conducting process.

Humans sense five basic tastes: sour, sweet, bitter, salty, and umami[1]. Among all the taste receptors, the receptor for sour taste remained elusive until a recent study demonstrated that OTOP1, a plasma membrane protein enriched in murine taste receptor cells, functions as an extracellular proton-activated proton channel and could potentially serve as the sour taste receptor[2]. Subsequent investigations in several animals further confirmed that OTOP1 is the bona fide sour taste receptor in mice and fruit flies[3–5]. Interestingly, before the discovery of its role in sour sensing, OTOP1 was initially identified as a key component in the formation of vestibular otoliths in mammals and

zebrafish. Mutations in OTOP1 can severely impact the body's balance and movement[6–8].

OTOP1 belongs to a protein family conserved from nematodes to humans[9]. Vertebrates contain three OTOP homologs (OTOP1-3) with distinct expression patterns[2]. In addition to the taste and vestibular cells, OTOP1 is also detected in brown adipose tissue, the mammary gland, and the heart[2,10]. In contrast, OTOP2 is expressed in the stomach, testis, and olfactory bulb, whereas OTOP3 is present in the epidermis, small intestine, stomach, and retina[11]. The physiological functions of OTOP2 and 3 have not yet been extensively explored and

[1]Howard Hughes Medical Institute and Department of Physiology, University of Texas Southwestern Medical Center, Dallas, TX, USA. [2]Department of Biophysics, University of Texas Southwestern Medical Center, Dallas, TX, USA. [3]Center for Life Sciences, Yunnan Key Laboratory of Cell Metabolism and Diseases, State Key Laboratory for Conservation and Utilization of Bio-Resources in Yunnan, School of Life Sciences, Yunnan University, Kunming, China. ✉e-mail: youxing.jiang@utsouthwestern.edu

remain largely unknown. Nevertheless, several studies have provided some suggestive insights into their potential functions. An immunostaining study involving human autopsy samples revealed the localization of OTOP2 in the saccular supporting cells of macula utricle[12], pointing to its involvement in otoconia formation and movement sensing in the vestibular system. Moreover, a notable reduction in OTOP2 expression has been observed in colorectal cancers[13], implying a possible contribution to cancer development. In sea urchins, the OTOP2 homolog exclusively presents in calcifying primary mesenchymal cells which generate the calcitic larval skeleton[14]. Interestingly, in egg-laying hens, the OTOP2 expression level in the uteri is elevated during the eggshell formation[15]. Remarkably, the nematode *Caenorhabditis elegans* contains eight Otopetrin-like proteins (CeOTOPs)[16] among which, CeOTOPs 1, 2, 4, and 7 are expressed in sensory neurons in the head of the worms, including the ASH and ASI neurons that are known for their sensitivity to acid stimulation and the rest four (CeOTOPs 3, 5, 6, and 8) are found in different parts of the pharynx. None of these CeOTOP channels has been functionally characterized and their physiological roles remain unclear; knockout of any single CeOTOP gene does not affect the proton currents of ASH neurons in response to acid stimuli[17].

Structures of several OTOP family members including chicken OTOP3, zebrafish OTOP1, and *Xenopus tropicalis* OTOP3 (XtOTOP3) have been determined, all of which show a dimerization of the channels with a similar double-barrel architecture within each subunit[18,19]. However, all these structures likely represent an inactive state and stop short of providing a clear picture of how proton conduction occurs in OTOP channels. Distinct from most multimeric channels that contain a central ion conduction pore, the large central hole enclosed by two OTOP subunits is occupied by lipid molecules and unlikely to be an ion-permeable pathway. In our earlier study, the double-barrel structural feature of OTOP led us to hypothesize that the two α-barrels within each subunit could both function as the proton-conducting pathway as each enclosed a cavity that is exposed to the extracellular side. However, the cavities in both barrels are sealed from the cytosolic side[18]. In addition to the cavity within each barrel, the interface between the two barrels also contains a cavity that is open to cytosol but sealed from the extracellular side. Molecular dynamics simulation suggested that the intra-subunit interface between the two barrels could also potentially serve as the proton-conducting pathway[19]. In addition to the unresolved proton-conducting pathway, the structural basis of extracellular pH activation and rapid desensitization in OTOP channels are also unresolved in these structures.

In our endeavor to reveal the structural mechanism of proton conduction in OTOP channels, we performed structural and functional characterization of the *Caenorhabditis elegans* OTOP8 (CeOTOP8) and mouse OTOP2 (mOTOP2), both of which can constitutively conduct proton at various extracellular pH[2,20,21]. By determining their structures at various pH, we revealed the conformational changes related to the proton conduction in OTOP and elucidated the bona fide ion-conducting pathway bridged by the conserved glutamate and histidine at the inter-barrel interface within each channel subunit. Combining all the available structures along with mutagenesis and functional analysis, we proposed a feasible working model for the proton-transferring process in OTOP channels.

## Results

### Functional analyses of CeOTOP8

The functional analyses were performed using patch clamp recordings of OTOP-expressing Human Embryonic Kidney 293 (HEK293) cells in whole-cell configuration. When the wild-type (WT) CeOTOP8 channel was expressed in HEK293 cells, no proton channel activity was observed in the recording (Fig. 1a). We initially attributed the lack of measurable current to the low expression of CeOTOP8 in HEK293 cells and designed a deletion construct (Δ1-57CeOTOP8) in which the

N-terminal 57 residues predicted to be disordered were removed. The expression of Δ1-57CeOTOP8 in HEK293 cells yielded large proton currents, allowing for functional characterization of CeOTOP8. Distinct from OTOP1 and OTOP3 channels that require acidic extracellular pH to elicit inward proton currents, Δ1-57CeOTOP8 conducts proton at various extracellular pH (Fig. 1a). Even with a symmetrical pH above neutral (i.e. pH 7.4), membrane voltage is sufficient to drive a large proton current (Fig. 1b). Similar to other OTOP channels, the proton currents of Δ1-57CeOTOP8 also decayed rapidly. As demonstrated in our previous study of XtOTOP3 and further discussed later, this current decay is caused by rapid local proton accumulation at the intracellular side upon channel activation, resulting in a pH-dependent inactivation of channel[18]. The pH gradient-driven inward proton currents plateau at an extracellular pH of around 5.5 (Fig. 1a), similar to that observed in mouse OTOP2 channel[20] (also see the later section). The proton conduction of the channel exhibits some inward rectification as demonstrated in the *I–V* curves obtained at various extracellular pH (Fig. 1c).

### Structures of full-length CeOTOP8

We initially targeted the functional CeOTOP8 construct for structural study as the channel is conductive even at high extracellular pH, which could potentially allow us to capture the channel in an activated state and provide structural insights into the proton-conducting process of the OTOP channel family. However, the functional construct was unexpectedly difficult to over-express and purify for structural analysis. On the contrary, the full-length CeOTOP8, albeit exhibits no channel activity in our recordings, could be easily overexpressed and purified for structural characterization. As demonstrated in our later structure-based mutagenesis study, the lack of channel activity in the WT CeOTOP8 is caused by self-inhibition from its N-terminal region rather than low expression. Serendipitously, this self-inhibition allows us to capture the WT CeOTOP8 channel structure in a transiently proton-transferring state as demonstrated in the next section, revealing the proton-conducting pathway in OTOP channels.

The WT CeOTOP8 was overexpressed in HEK293F cells and purified in glyco-diosgenin (GDN) detergent. Its structures were determined at pH 5.0 and 8.0 using single-particle cryo-EM (Materials and Methods and Supplementary Figs. 1–4). The overall structural arrangement of CeOTOP8 is similar to other OTOP channel structures. It forms a dimer with each subunit containing 12 transmembrane (TM) helices (Fig. 2a, b). The TM region of each channel subunit can be divided into two structurally homologous halves that assemble into a double-barrel architecture (Fig. 2b, c). Five TMs from each half (TMs 2–6 in N-half or TMs 8–12 in C-half) form an individual α-helical barrel and the first TM of each half (TM1 or TM7) is positioned on the periphery. TM1 participates in the dimerization interaction with TMs 9&10 of the neighboring subunit (Fig. 2b, d). Notably, CeOTOP8 contains three unique structural elements that were not observed in any other OTOP channel structures (Fig. 2b): its N-terminal loop preceding TM1 (residues 50–57) is well structured and provides additional dimerization interaction with TMs 8 and 10 of the neighboring subunit at their cytosolic ends (Fig. 2e); post-TM5 residues (244–270) form an extended helix in the high pH structure, but remain disordered in the low pH structure; CeOTOP8 has a much longer protein sequence between TMs 2 and 3 and part of it form a helix (α2) that sits parallel to the intracellular membrane surface and protrudes away from the channel dimer (Fig. 2b).

Intriguingly, the N-terminal structured loop from residues 50–57 is deleted in the functional Δ1-57CeOTOP8 construct. To test if these residues are important for the activity of CeOTOP8, we generated two deletion mutants of CeOTOP8 in which the first 49 residues (Δ1-49CeOTOP8) or just residues 50–57 (Δ50-57CeOTOP8) were removed, respectively. The first deletion construct which retains the structured loop has no channel activity similar to the full-length channel whereas the second deletion construct exhibits high channel activity, confirming the inhibitory role of the N-terminal structured loop in CeOTOP8

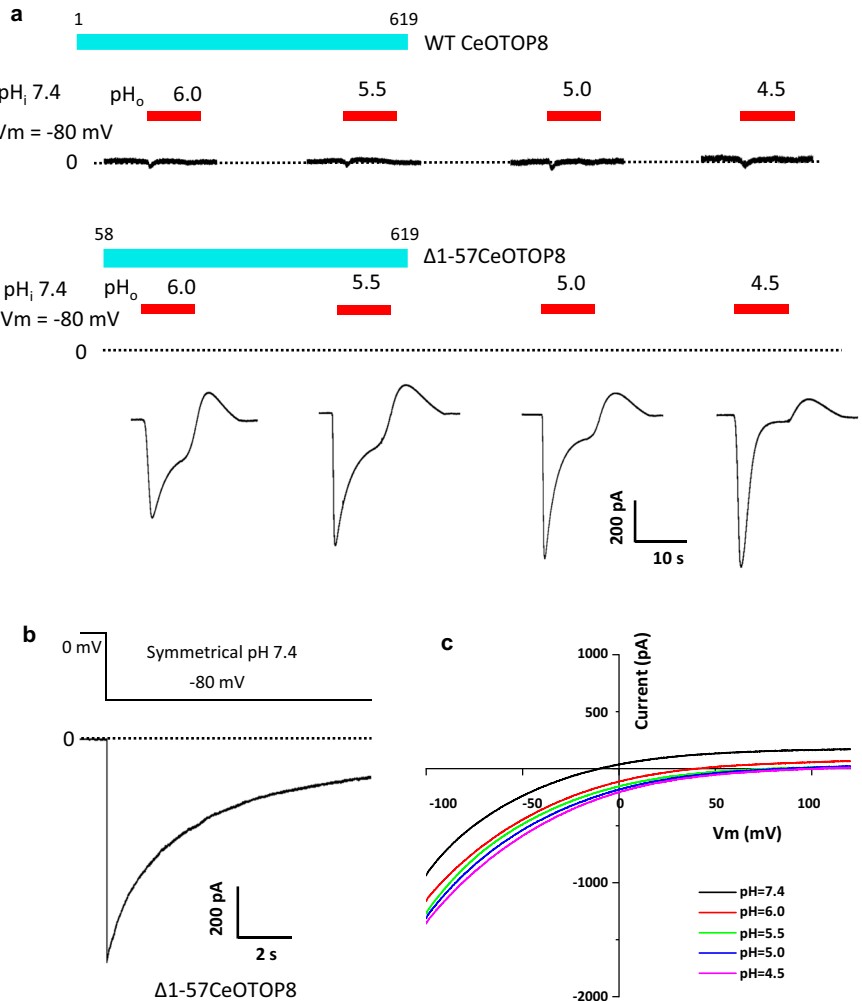

**Fig. 1 | Functional characterization of CeOTOP8. a** Sample traces of whole-cell recordings of HEK293 cells expressing the wild-type CeOTOP8 (top) and the Δ1-57CeOTOP8 mutant (bottom). The inward proton currents were elicited by changing the extracellular pH (pHo) from 7.4 to various acidic pH with −80 mV holding potential. **b** A sample trace of whole-cell inward proton current measured at −80 mV with a symmetrical pH of 7.4. **c** Sample *I–V* curves of Δ1-57CeOTOP8 with varying extracellular pH. The intracellular pH is set at 7.4.

activity (Fig. 2f). Within this region, Trp54 appears to be the main residue that engages in the dimerization interaction (Fig. 2e). To our surprise, W54A single mutation is sufficient to revive the channel activity of CeOTOP8 (Fig. 2f), indicating that the Trp54-mediated dimerization interaction plays the key role in channel inhibition. As further discussed in the following study of mouse OTOP2, the C-barrel of the OTOP channel likely undergoes a *pendulum clock-like* sliding movement relative to the TM1 from the neighboring subunit during proton conduction and therefore the inter-subunit movement at the dimer interface would impact the channel activity. We suspect that the interaction between Trp54 and the C-barrel of the neighboring subunit confines this sliding movement in CeOTOP8 and thereby inhibits the channel activity.

## CeOTOP8 structures represent two distinct states during proton conduction

The CeOTOP8 structures determined at pH 5.0 and 8.0 exhibit some key differences at the inter-barrel interface, particularly near the two absolutely conserved interfacial residues of Glu325 and His567, and represent two distinct states during proton conduction.

At low pH, the inter-barrel interface of CeOTOP8 resembles those in OTOP1 and OTOP3 structures where the conserved glutamate (Glu325 in CeOTOP8) is buried at the center of the hydrophobic interface between the two barrels (Fig. 3a, b). As pKa of an acidic residue can

increase significantly when buried in a hydrophobic environment[22–24], this glutamate is likely protonated in the low pH CeOTOP8 structure. Previous mutagenesis studies of XtOTOP3 demonstrated that this glutamate could be replaced by Asp without affecting channel activity, but its neutralization mutation to Gln completely abolished the channel activity[18]. The mutagenesis results, along with its conservation and intriguing localization, indicate an essential role of this interfacial glutamate in OTOP function. His567 of CeOTOP8, another highly conserved interfacial residue, is positioned at the bottom of a deep and intracellular solvent-accessible cavity formed between the two barrels (Fig. 3a). In light of low pH condition, His567 is expected to be protonated and forms a salt bridge interaction with Asp439 on TM9. Similar to the above-mentioned glutamate, this histidine also plays a central role in OTOP proton conduction and its mutations to Ala or Gln led to the complete loss of function in XtOTOP3[18], a similar phenotype was also observed in zebrafish OTOP1[19].

At high pH, the CeOTOP8 inter-barrel interface adopts an intriguing structure. The Glu325 side chain points down towards the intracellular side whereas the His567 imidazole ring adopts two alternative conformations (Fig. 3a, c). The His567 side chain flips up toward the extracellular side in its major conformation, forming a salt bridge with Glu325 (Fig. 3a, c). This structure likely represents a transient state in which the protonated Glu325 is transferring its proton to His567. In

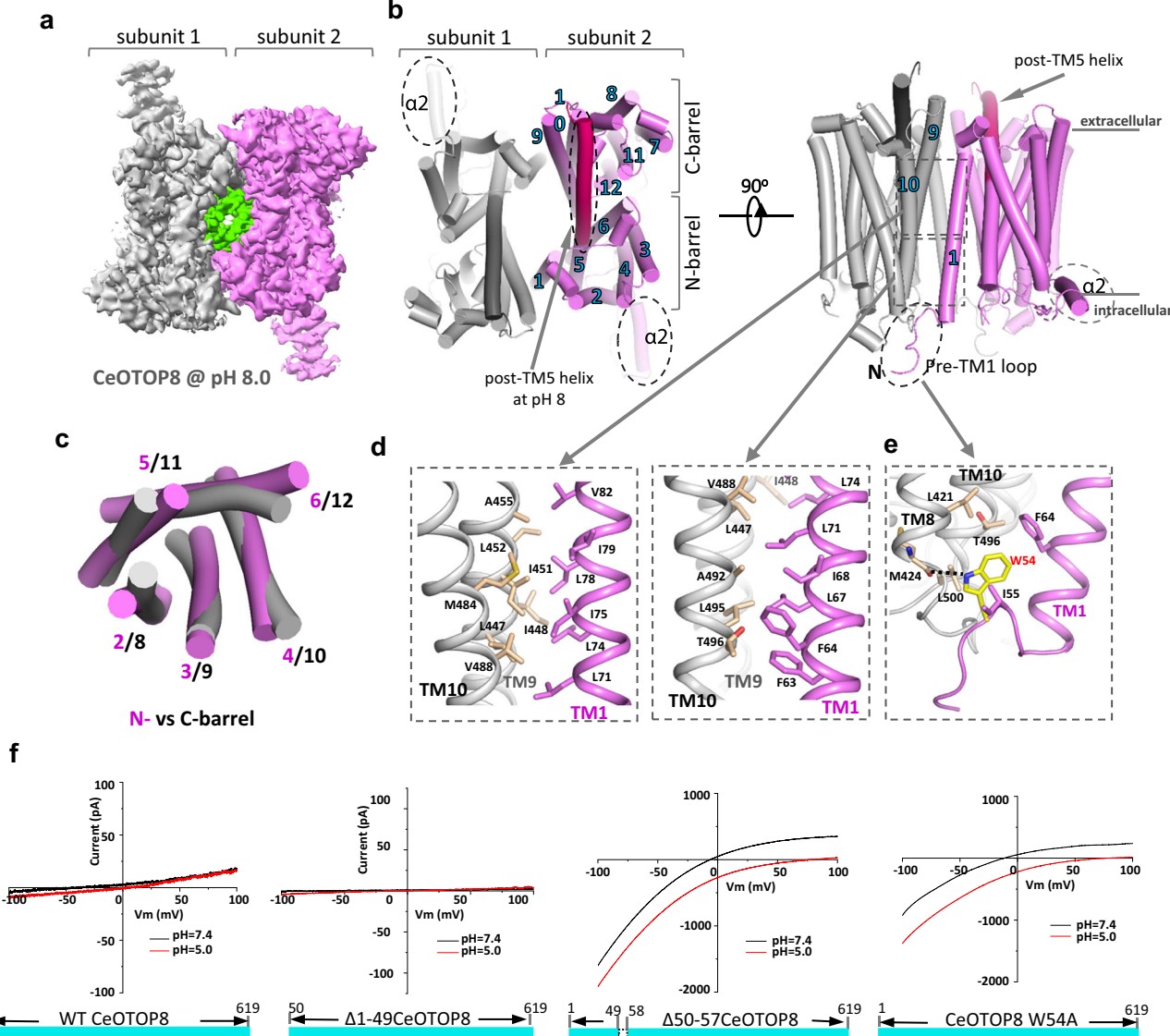

**Fig. 2 | The overall structure of CeOTOP8 and the structural basis of its self-inhibition. a** Top view of a 3D reconstruction of CeOTOP8 dimer at pH 8.0 with each subunit individually colored and lipid density shown in green. **b** Cylinder representation of the CeOTOP8 dimer structure at pH 8.0 in top and side views. Numbers mark the TM helices of one subunit. Dotted ovals enclose the structural features unique in CeOTOP8. The extended TM5 helix at pH 8 is colored in hot pink. **c** Superimposition between the N-barrel (TMs 2–6 in violet) and C-barrel (TMs 8–12 in grey) within each subunit. **d** Zoomed-in view of the inter-subunit dimerization contacts between TMs 9 and 10 from subunit 1 (grey) and TM1 from subunit 2 (violet). The view is divided into two segments marked by the dotted boxes in (**b**). **e** A zoomed-in view of the interaction between the N-terminal loop and residues from TMs 8 & 10 of the neighboring subunit. The dotted line marks the hydrogen bond between W54 and the carbonyl of M424. **f** Constructs and representative *I–V* curves of WT CeOTOP8 and its mutants. The recordings were performed at an extracellular pH of 7.4 or 5.0 and an intracellular pH of 7.4.

its minor conformation, the His567 side chain remains downward-facing in the cavity and exposed to the cytosolic solution. In addition to the side-chain movements at Glu325 and His567, two other major structural changes also occur in CeOTOP8 from low to high pH (Fig. 3d, e). Firstly, TM12 of CeOTOP8 forms a bent helix at low pH with a kink at a highly conserved proline (Pro559) residue, whereas the N-terminal part of the TM12 helix before Pro559 unwinds to a loop at high pH (Fig. 3e). This helix-to-loop structural transition allows the Phe563 side chain to rotate upward, vacating the space necessary to accommodate the upward flip of the His567 imidazole as well as the downward rotation of Glu325 side chain (Fig. 3d, e and Supplementary Movie 1). Thus, this structural transition appears to be prerequisite for the formation of the proton-transferring bridge between Glu325 and His567. The other structural change is that the residues 244–270 after the C-terminus of TM5 are disordered at low pH but form a helix that

extends seamlessly after TM5 at high pH (Fig. 3d). This extended helix leans over the top of the C-barrel, pushing the loop between TM11 and TM12 towards the center of the barrel and may facilitate the helix-to-loop transition at TM12. However, the amino-acid sequence and length after TM5 vary among OTOP channels (Supplementary Fig. 4) and the exceptionally long post-TM5 helix observed at higher pH is likely unique for CeOTOP8.

## Structural and functional characterization of mouse OTOP2

While CeOTOP8 was our initial focus on studying the proton conduction of OTOP channels, we also adopted the mouse OTOP2 (mOTOP2) as an alternative model system in our structural approach because mOTOP2 exhibits a similar constitutive proton-conducting activity at various extracellular pH[2,20,21] (Fig. 4). At symmetrical pH 7.4, the negative membrane potential is sufficient to elicit proton currents and

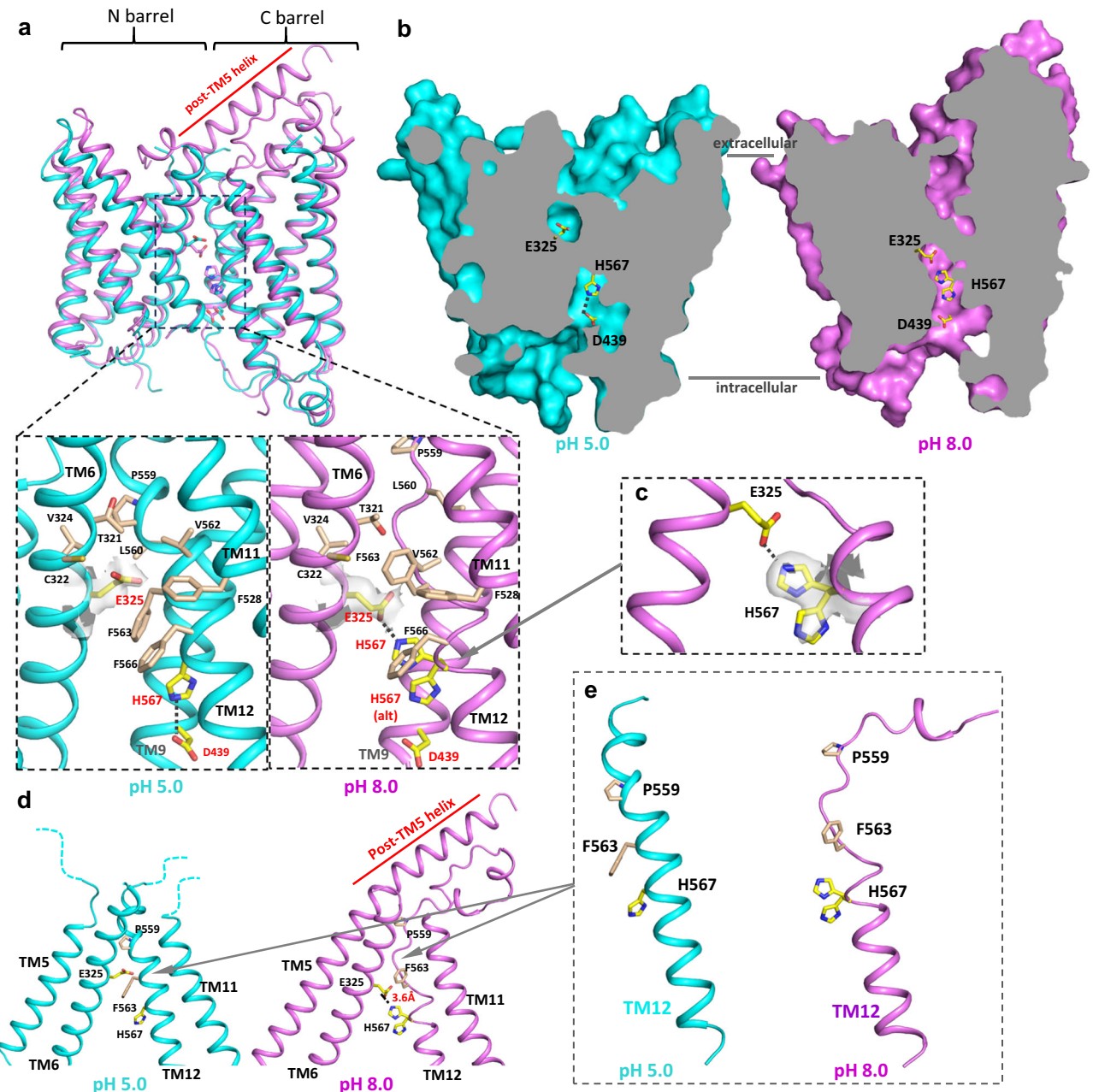

**Fig. 3 | Comparison of CeOTOP8 structures at low and high pH.**
**a** Superimposition of CeOTOP8 structures obtained at pH 5.0 (cyan) and pH 8.0 (violet) with zoomed-in views of the surrounding environment of E325 at the inter-barrel interface. TMs 1 & 7 and α2 are removed in comparison for clarity. Conserved Glu325, Asp439 and His567 are labeled in red. His567 (alt) marks its alternative conformation. EM densities of Glu325 are shown in grey surface contoured at 3 σ.

**b** Cross sections of the surface-rendered CeOTOP8 structures show that E325 is buried at the interface in low pH but forms a salt bridge with H567 in high pH. **c** EM density of H567 (grey surface contoured at 5 σ) shows the two alternative conformations of its side chain. **d** Two major structural changes at TM5 and TM12 of CeOTOP8 from low to high pH. **e** Helix formation and unwinding at the N-terminal part of TM12 in low and high pH, respectively.

the channel quickly desensitizes as seen in other OTOP channels (Fig. 4a). In the recordings of channel activity at various acidic extracellular pH (both at 0 and −80 mV holding potentials), the inward proton currents driven by pH gradient start to plateau at an extracellular pH of about 5.5, suggesting the presence of a rate-limiting step during proton conduction independent of extracellular proton concentration. A similar observation was also made in a recent functional characterization of mOTOP2 expressed in the *Xenopus* oocytes[2]. The I−V curves of the channel obtained at various extracellular pH are almost parallel to each other with no obvious rectification and have pH gradient-correlated reversal potentials (Fig. 4c), confirming the proton selectivity as well as the lack of external acid activation in mOTOP2.

Mouse OTOP2 was expressed in HEK293F cells and purified in glyco-diosgenin (GDN) detergent. Its single-particle cryo-EM structures were determined using protein samples prepared at various pH conditions (5.0, 7.0, 8.0, and 9.0) (Materials and Methods and Supplementary Figs. 5–9). The mOTOP2 structures at pH 8.0 and 9.0 are virtually identical. The mOTOP2 protein becomes more dynamic at pH 7.0 and its particles can be classified into two groups. The structure from the major group of particles (3.22 Å) is similar to that determined at pH 5.0 (3.06 Å). Albert at a lower resolution, the mOTOP2 structure obtained from the minor group of particles (3.79 Å, Method and Supplementary Fig. 7) adopts a conformation different from those obtained at low and high pH and will be further described in the later

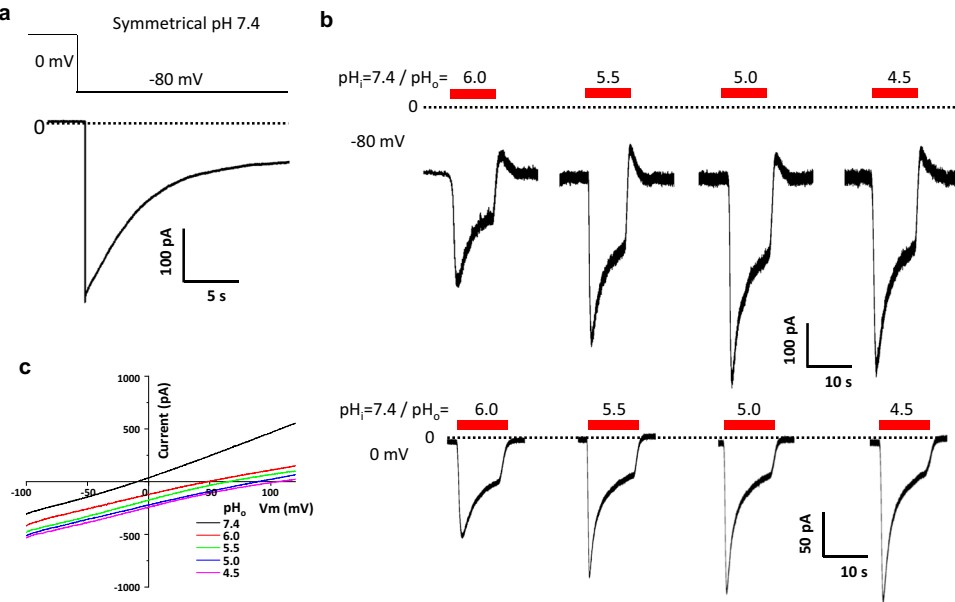

**Fig. 4 | Functional characterization of mouse OTOP2. a** A sample trace of whole-cell recordings of HEK293 cells expressing the wild-type mOTOP2. The inward proton current was measured at −80 mV with a symmetrical pH of 7.4. **b** Sample traces of whole-cell inward proton current measured at −80 and 0 mV. The currents were elicited by switching the extracellular pH (pHo in the bath) from 7.4 to various acidic pHs. The intracellular pH (pHi in the pipette) is set at 7.4. **c** Sample I–V curves of mOTOP2 with varying extracellular pH conditions. The intracellular pH is set at 7.4.

discussion about inter-subunit conformational change. Among all, the mOTOP2 structures determined at pH 5.0 and 8.0 are better resolved and will be used to represent the low and high pH structures, respectively, in the following discussion.

Same as other OTOP channels, mOTOP2 forms a dimer with each subunit containing 12 transmembrane (TM) helices that assemble into two parallel and structurally homologous helical barrels (Fig. 5a). The structure difference in mOTOP2 between low and high pH conditions is quite subtle and mainly occurs at the N-terminal half of TM6 preceding the highly conserved glutamate (Glu250 in mOTOP2) important for proton transfer (Fig. 5b). At low pH, TM6 of mOTOP2 forms a bent helix with a kink at Pro246 just one helical turn above Glu250 whose side chain, likely in a protonated state, is pinned to the center of the hydrophobic interface between the N- and C-barrels (Fig. 5b, c, left insets). The same kinked TM6 was also observed in the previous OTOP1 and 3 structures[18,19]. At higher pH, however, the N-terminal half of the TM6 helix unwinds to an extended loop (Fig. 5b, c, right insets). This helix-to-loop structural transition occurs before the conserved Pro246 and vacates the space above Glu250 initially occupied by Pro246, allowing the Glu250 side chain to flip upwards and become completely exposed to the external solution (Fig. 5c, d). The outward-facing Glu250 is also stabilized by forming an H-bond with Tyr245 and a salt bridge with Arg517 in the high pH mOTOP2 structure (Fig. 5c). As the conformational change at TM6 determines the accessibility of Glu250 to external solvent, the two structures obtained at different pH likely represent two distinct states during proton conduction in OTOP channels: the high pH one represents a state in which the central glutamate is exposed to external solution for protonation; the low pH one represents a state in which the protonated glutamate is buried in the inter-barrel interface and ready to transfer its proton to the histidine acceptor, similar to the low pH structure of CeOTOP8.

**Implication of intra- and inter-subunit conformational changes during proton conduction**

In the CeOTOP8 structures, the proton-transferring Glu325 and His567 residues are positioned in close proximity both at high and low pH with

Cβ-to-Cβ distances of about 8.6 and 8.8 Å, respectively, allowing them to form a salt bridge by simple side-chain rotation. In mOTOP2, however, the two equivalent residues (Glu250 and His551) are too far apart (with Cβ-to-Cβ distances of about 14.8 Å at high pH and 12.8 Å at low pH) to form a similar proton-transferring bridge by a simple side-chain movement (Fig. 6a), suggesting that conformational change is required to move them closer before they can engage in proton transfer. When mOTOP2 and CeOTOP8 are superimposed at the N-barrel, their C-barrels show a swing movement hinged at an external edge of the C-barrel near the loop between TMs 9 and 10 (Fig. 6b). Applying this C-barrel swing movement in mOTOP2 would move its His551 closer to Glu250 for salt bridge formation. Thus, structural comparison between mOTOP2 and CeOTOP8 implies an inter-barrel swing motion within each subunit that controls the distance between the two proton-transferring residues and thereby regulates channel activation.

In the context of the OTOP channel dimer, the C-barrel swing motion would result in an inter-subunit *pendulum clock-like* slide between the C-barrel and TM1 from the neighboring subunit (Fig. 6c). As the C-barrel movement is hinged at the external ends of the two TM helices (TMs 9 and 10) that are engaged in direct dimerization interaction with TM1, the relative movement between TM1 and C-barrel at the dimer interface becomes progressively larger from extracellular to intracellular side (Fig. 6c). This inter-subunit sliding motion appears to be central to defining the C-barrel position for proton transfer and therefore how easy the two subunits can slide relative to each other at the dimer interface would directly affect the OTOP channel activity. Consistent with this hypothesis, the inter-subunit interaction between Trp54 right before TM1 and the C-barrel at the cytosolic end of the CeOTOP8 dimer interface constrains the C-barrel movement and thereby inhibits the channel activity; releasing this constrain by a simple W54A mutation is sufficient to yield a highly active channel (Fig. 2e, f). It is worth noting that in the pH 7 mOTOP2 structure obtained from the minor group of particles, its C-barrel moves to an intermediate state between the two main C-barrel conformations observed in mOTOP2 and CeOTOP8, confirming the dynamic motion of the C-barrel in mOTOP2 (Supplementary Fig. 7b).

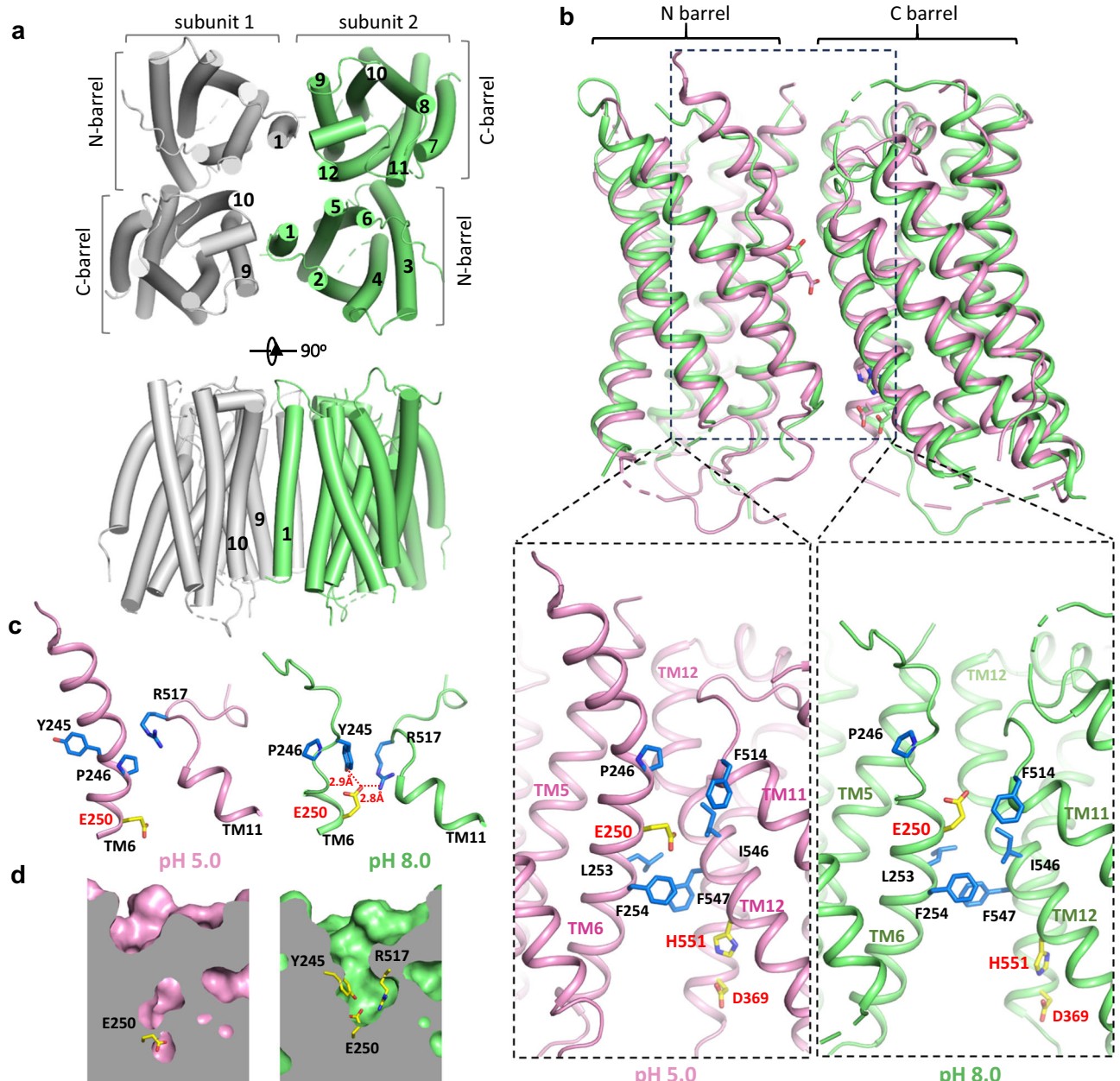

**Fig. 5 | Structures of mOTOP2. a** Cylinder representation of the overall mOTOP2 dimer structure at pH 8.0 in top and side views. Numbers mark the TM helices. **b** Superimposition of mOTOP2 structures obtained at pH 5.0 (pink) and pH 8.0 (green) with zoomed-in views of the surrounding environment of E250 at the inter-barrel interface. The conserved Glu250, Asp369, and His551 are labeled in red. **c** Zoomed-in view of the helix-to-loop conformational change at TM6 from pH 5.0 to pH 8.0. **d** Cross sections of the surface-rendered mOTOP2 structures show that E250 is buried at the interface in low pH but exposed to the extracellular solution at high pH.

To verify the importance of inter-subunit contact for channel activity, we performed mutagenesis at the dimer interface of mOTOP2. In these mutations, one or two interfacial residues on TM1 or TM9 are replaced by tryptophan (Fig. 6d). Among all tested, several single or double mutants remain functional, including L34W, N38W, T45W/L46W, and L41W/L42W (Fig. 6e). No channel activity was observed in three single mutants, including L30W, V370W, and M374W (Fig. 6e). The lack of channel activity in these mutants are not caused by low expression in HEK cells or the disruption of their dimerization, as all three loss-of-function mutants were expressed at an even higher level than the wild-type channel and all can be purified as a stable dimer (Supplementary Fig. 10a, b). Furthermore, we also determined the cryo-EM structure of one of the loss-of-function mutants, M374W, at

pH 8.0 and demonstrated that other than a different side chain at residue 374 the mutant structure is virtually identical to the wild-type mOTOP2 (Supplementary Fig. 10c, d; Supplementary Fig. 11). It is interesting to note that those functional mutations are largely clustered at the extracellular half of the interface whereas the three loss-of-function mutations are all positioned near the intracellular end of the interface. The functional effects of these interfacial mutations correlate with the structural observation of progressively increased sliding movement towards the intracellular side of the dimer interface between TM1 and C-barrel. We suspect that the interfacial Trp-scanning mutagenesis changes the propensity of the inter-subunit movement and those mutations near the cytosolic side of the interface inflict stronger hindrance to the interfacial sliding movement and

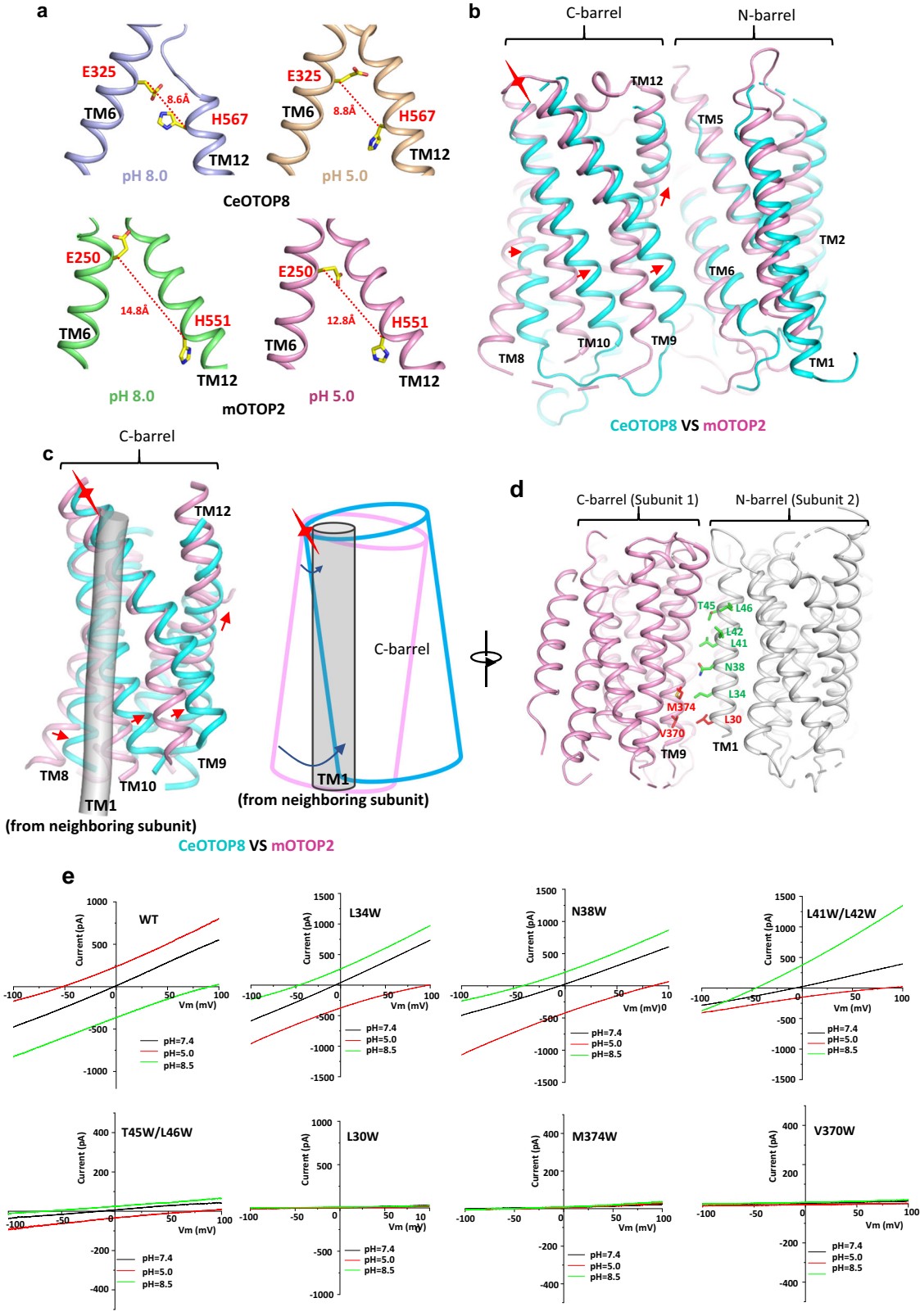

mitigate the channel activity, reminiscent of the inhibitory effect of Trp54 at the N-terminus of TM1 in CeOTOP8 channel.

## Discussion

Combining the structural studies of both mOTOP2 and CeOTOP8 at various pH, we identified three key conformational changes surrounding the proton-carrying glutamate and histidine at the inter-barrel interface that are important for proton conduction in OTOP channels. Firstly, the helix-to-loop transition at the N-terminal part of TM6 observed in mOTOP2 determines whether the glutamate is exposed to the extracellular solution for protonation or buried in the inter-barrel interface in a protonated state ready to contribute its proton to an acceptor. Secondly, the helix-to-loop transition at the N-terminus of TM12 observed in CeOTOP8 and the subsequent

**Fig. 6 | Intra- and inter-subunit conformational changes. a** Distances between the proton-transferring glutamate and histidine residues in CeOTOP8 (top) and mOTOP2 (bottom). The distances are measured between their Cβ atoms from structures obtained at pH 5.0 and 8.0. **b** Intra-subunit conformational changes implicated from the structural comparison between CeOTOP8 (cyan) and mOTOP2 (pink). The low pH structures are used in the superimposition. Red arrows mark the swing motion of the C-barrel, and the red 4-point star marks the hinge at the loop between TMs 9 and 10. **c** Inter-subunit conformational changes between C-barrers and TM1 implicated from the structural comparison between the channel dimers of

CeOTOP8 (cyan) and mOTOP2 (pink). For simplicity, only TM1 of CeOTOP8 (grey cylinder) is shown in the comparison. Red arrows mark the direction of the movement. The cartoon drawing depicts the *pendulum clock-like* slide between the C-barrel and TM1, illustrating the progressively increased relative movement from the extracellular to the intracellular side. **d** Positions of those dimer-interface amino acids on TM1 or TM9 targeted for tryptophan-scanning mutagenesis. Residues with functional mutations are labeled in green and non-functional mutations in red. **e** Sample $I–V$ curves of WT mOTOP2 and its mutants recorded with extracellular pH of 8.5, 7.4, and 5.0. The intracellular pH is set at 7.4.

movement of Phe563 vacate the space necessary for the salt bridge formation between Glu325 and His567. Consequently, the un-protonated histidine can flip its imidazole ring upward and serve as a proton acceptor by forming a salt bridge with the protonated Glu325. Thus, we could visualize the proton-conducting pathway in an OTOP channel bridged by the highly conserved glutamate and histidine at the inter-barrel interface. During proton conduction in CeOTOP8, we expect that the proton transfers from Glu325 to His567, followed by the downward flip of His side chain and subsequent release of its proton to the cytosol. Indeed, in the high pH structure, the density map at His567 clearly shows two alternating side chain conformations with one in upward and the other in downward positions (Fig. 3c). Thirdly, the relative swing motion between N- and C-barrel within each subunit determines whether the glutamate and histidine residues are close enough to form a proton-transferring bridge. In light of channel dimerization, this swing motion entails a sliding movement between the two channel subunits, and therefore mutations at the dimer interface can have a significant impact on channel activity.

Thus, unlike conventional channels, OTOP channels do not have a particular conformation representing a conductive open state. Instead, the proton conduction process of OTOP channels involves multiple steps. The mOTOP2 and CeOTOP8 structures provide snapshots of the OTOP channel in various states, enabling us to propose a working model describing the proton conduction process (Fig. 7). In state 1, a loop configuration at the N-terminal part of TM6 allows the glutamate to be exposed to the extracellular solution for protonation. Once protonated, the loop-to-helix transition at the N-terminus of TM6 drives the proton-carrying glutamate side chain to the middle of the inter-barrel interface as depicted in state 2. The two conformations observed in mOTOP2 structures represent these first two states of the proton-conducting process, whereas the CeOTOP8 structures likely capture the next two states (states 3 and 4). In state 3, the C-barrel swings upward to move the histidine close enough to the glutamate for proton transfer. In state 4, the helix-to-loop transition at the N-terminus of TM12 vacates the space necessary for the histidine to flip its side chain upward and form a proton-transferring salt bridge with the protonated glutamate. Upon accepting the proton, the protonated histidine flips its side chain back into the cytosol (state 5), followed by the release of its proton into the cytosolic solution (state 6). Following the histidine movement in state 5, the N-terminus of TM12 must reform the helix along with the downward swing of the C-barrel so that the glutamate can flip its side chain position back in the inter-barrel interface and get ready to be re-protonated from the extracellular side (state 6). However, the negatively charged glutamate becomes unstable at the hydrophobic inter-barrel interface. Consequently, the helix-to-loop transition at TM6 exposes the glutamate to the external solution, and the channel cycles back to the conformation in state 1. While our working model only depicts the proton influx process, the proton conduction is reversible depending on the electrochemical driving force and internal/external pH environment. The depicted multi-step transporter-like proton conduction process in OTOP channels implies that the rate of conformational changes in the cycle would determine the proton flux rate. Indeed, the proton currents of both CeOTP8 and mOTOP2 channels become saturated once the extracellular pH reaches about 5.5 (Figs. 1a and 4b), suggesting the presence

of a rate-limiting step during proton conduction independent of proton concentration.

The central role of the conserved intracellular histidine (His567 in CeOTOP8 and His551 in mOTOP2) as a proton acceptor along with its location deep in the narrow cavity provides a plausible explanation for the OTOP channel desensitization caused by rapid proton influx. Positioned in a confined cavity space where ion diffusion is expected to be slowed, the His-mediated proton influx would cause a local proton accumulation, resulting in a lower apparent pH in the cavity than that of the cytosolic solution. This local proton accumulation prolongs the protonation state of the histidine even with higher cytosolic pH, thereby preventing it from accepting the proton from glutamate and stalling the proton conduction of the channel. Three functional properties observed in our previous study of XtOTOP3 are consistent with this desensitization mechanism[18]. Firstly, the desensitization rate of the OTOP channel depends on the proton influx rate, a higher influx causes a faster desensitization. Secondly, an intracellular pH of about 6.0 is sufficient to inhibit the proton influx of XtOTOP3 likely caused by the histidine protonation. Thirdly, when applying repetitive voltage ramps from −100 mV to 100 mV in the recording of XtOTOP3 with an inward proton gradient ($pH_i = 7.4$ and $pH_o = 5.0$), there was a progressive decrease in reversal potential as if the intracellular pH was progressively decreased, consistent with a local proton accumulation. Furthermore, when facing inside, the protonated histidine is stabilized by forming a salt bridge with a conserved aspartate on TM9 in OTOP channels (Asp439 in CeOTOP8, Asp369 in mOTOP2, and Asp509 in XtOTOP3), which effectively increases the apparent pKa of the histidine and makes its de-protonation more difficult. In other words, this His-Asp salt bridge interaction would hinder the proton release from the histidine and aggravate the channel desensitization. Indeed, mutations of Asp509 to Asn or Ala in XtOTOP3 can markedly reduce the desensitization rate[18].

## Methods

### Protein expression and purification

The *Caenorhabditis elegans* Otopetrin-like 8 (CeOTOP8) gene with a C-terminal thrombin cleavage site and a 10× His tag and the *Mus musculus* Otopetrin 2 (mOTOP2) gene with a N-terminal 10× His tag followed by Green Fluorescent Protein (GFP) and a thrombin cleavage site were cloned into a pEZTBM vector[25] and heterologously expressed in HEK293F cells (ATCC cat# CRL-3022) using the BacMam system. The baculovirus was produced in *Spodoptera frugiperda* 9 (Sf9, Gibco cat#11496015) cells and used to transduce the HEK293F cells at a ratio of 1:40 (virus:HEK293F, v/v) and supplemented with 1 mM sodium butyrate to boost the protein expression. We failed to generate the gain-of-function Δ1-57CeOTOP8 truncation mutant virus following the same protocol, even after extensive trials using different tags at both N- and C-terminus. Sf9 cells transfected with the mutant bacmid became sick, likely because of the potent proton conduction activity of the chimera. Cells were cultured in suspension at 37 °C for 48 h and harvested by centrifugation at 3000 g. All purification procedures were carried out at 4 °C unless specified otherwise. The cell pellet was resuspended in buffer A (20 mM Tris pH 8.0, 150 mM NaCl) supplemented with a protease inhibitor cocktail (containing 1 mg ml⁻¹ each of DNase, pepstatin, leupeptin, and aprotinin and 1 mM PMSF) and

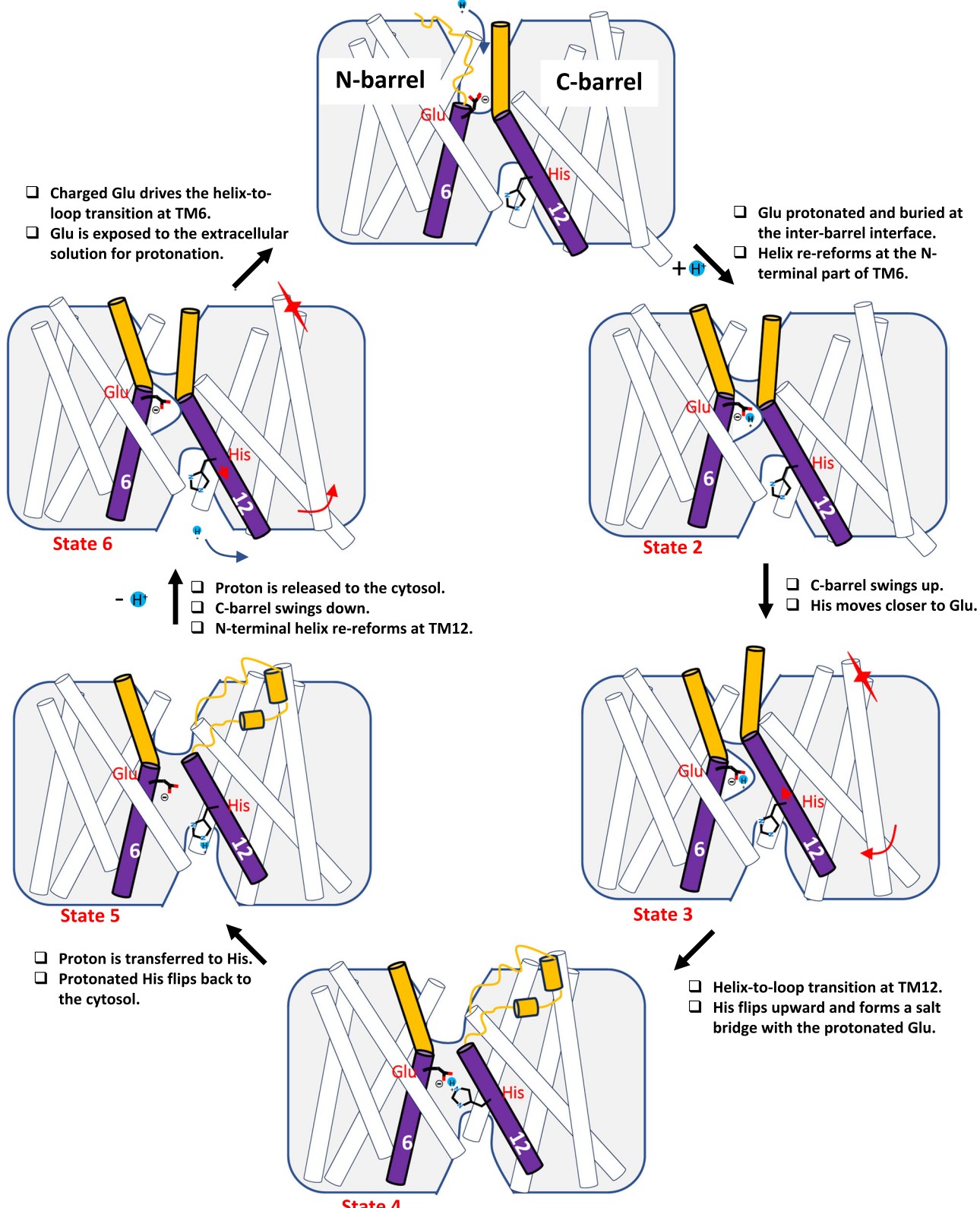

**Fig. 7 | Proposed working model of inward proton conduction in OTOP channels.** The conformational changes are centered around the conserved glutamate on TM6 and the histidine on TM12 at the inter-barrel interface.

homogenized by sonication on ice. Protein was extracted with 1% (w/v) n-dodecyl-β-D-maltopyranoside (DDM; Anatrace) supplemented with 0.2% (w/v) cholesteryl hemisuccinate (CHS; Sigma-Aldrich) by gentle agitation for 2 h. After extraction, the supernatant was collected after a

1 h centrifugation at 48,000 × g and incubated with Ni-NTA resin and 20 mM imidazole with gentle agitation. After 1 h, the resin was collected on a disposable gravity column (Bio-Rad), washed with buffer B (buffer A + 0.04% glyco-diosgenin (GDN; Anatrace)) with 20 mM

imidazole. The washed resin was left on-column in buffer B and digested with thrombin overnight. After digestion, the flow-through was concentrated, and purified by size-exclusion chromatography on a Superose 6 10/300 GL column (GE Healthcare) pre-equilibrated with buffer B. The protein peak was collected and concentrated to a final concentration of 3.0 mg/mL. For the sample at pH 5.0, an appropriate volume of acetic acid stock (1%) was added to the purified protein. To determine the amount of acetic acid needed to adjust the pH of the protein sample, a titration was performed using a larger volume of the buffer solution (with 20 mM Tris buffer at pH 8) used for protein purification. When titrated with 50 mL of buffer solution, 30 μL of pure glacial acetic acid is needed to achieve pH 5, equivalent to a 0.06% (v/v) final concentration. Thus, the protein sample at pH 5.0 was obtained by adding acetic acid to a 0.06% (v/v) final concentration to the purified protein before grid preparation.

### Electron microscopy data acquisition

The cryo-EM grids were prepared by applying 3.5 μl of purified protein (3.0 mg/mL) to a glow-discharged Quantifoil R1.2/1.3 300-mesh gold holey carbon grid (Quantifoil, Micro Tools GmbH) and blotted for 3.0 s under 100% humidity at 4 °C before being plunged into liquid ethane using a Mark IV Vitrobot (FEI). Micrographs were acquired on a Titan Krios microscope (FEI) operated at 300 kV with a K3 Summit direct electron detector (Gatan), using a slit width of 20 eV on a GIF-Quantum energy filter. Data were collected using CDS (Correlated Double Sampling) mode of the K3 camera with a super-resolution pixel size of 0.415 Å. The defocus range was set from −0.9 to −2.2 μm. Each movie was dose-fractionated to 60 frames with a dose rate of $1e^-/Å^2$/frame for a total dose of $60e^-/Å^2$. The total exposure time was between 5 to 6 s. Micrographs of mOTOP2 M374W dataset were acquired on a Titan Krios microscope (FEI) operated at 300 kV with a Falcon 4 electron detector (Thermo Fisher), using a slit width of 20 eV on a post-column Selectris X energy filter (Thermo Fisher Scientific). Data was collected using Falcon 4 camera with a pixel size of 0.737 Å. The defocus range was set from −0.9 to −2.2 μm. Each movie was dose-fractionated to 63 frames with a dose rate of $0.957e^-/Å^2$/frame for a total dose of $60e^-/Å^2$. The total exposure time was between 4 to 5 s.

### Image processing

Movie frames were motion-corrected and dose-weighted using MotionCor2[26]. The CTF parameters of the micrographs were estimated using the GCTF program[27]. The rest of the image processing steps were carried out using RELION 3.1[28–30]. All resolution was reported according to the gold-standard Fourier shell correlation (FSC) using the 0.143 criterion[31]. Local resolution was estimated using RELION. All micrographs were manually inspected to remove those with ice contamination and bad defocus. Particles were selected using Gautomatch (K. Zhang, MRC LMB, https://www2.mrc-lmb.cam.ac.uk/research/locally-developed-software/zhang-software/) and extracted with a binning factor of 3. 2D classification was performed in RELION 3.1. Selected particles after 2D classification were subjected to one round of 3D classification. The map from *Xenopus tropicalis* OTOP3 (XtOTOP3) (EMD-0650[18]) was low-pass filtered to 30 Å and used as the initial reference. Classes with clear OTOP channel features were combined and subjected to 3D auto-refinement and another round of 3D classification without performing particle alignment using a soft mask around the protein portion of the density. The particles of best-resolving classes were then re-extracted with the original pixel size and further refined. Beam tilt, anisotropic magnification, per-particle CTF estimations, and Bayesian polishing were performed in RELION to improve the resolution of the final reconstruction.

For the dataset of CeOTOP8 at pH 5.0, 5151 movies were collected, and 5070 movies were selected after motion correction and CTF estimation. 1,261,211 particles were extracted from the selected micrographs and subjected to one round of 2D classification, from which 727,678 particles were selected. After the initial 3D classification, 186,993 particles were selected and subjected to 3D auto-refinement. Next, a soft mask excluding the micelle density was applied and particles were classified into 5 classes without performing alignment. From this, one class with 41,311 particles was selected and further refined, yielding a map at 2.91 Å overall resolution (Supplementary Fig. 1).

For the dataset of CeOTOP8 at pH 8.0, 5481 movies were collected, and 5239 movies were selected after motion correction and CTF estimation. 1,157,107 particles were extracted from the selected micrographs and subjected to one round of 2D classification, from which 553,883 particles were selected. After the initial 3D classification, 130,051 particles were selected and subjected to 3D auto-refinement. Next, a soft mask excluding the micelle density was applied and particles were classified into 5 classes without performing alignment. From this, one class with 48,261 particles was selected and further refined, yielding a map at 3.02 Å overall resolution (Supplementary Fig. 2).

For the dataset of mOTOP2 at pH 5.0, 4162 movies were collected, and 4095 movies were selected after motion correction and CTF estimation. 967,811 particles were extracted from the selected micrographs and subjected to one round of 2D classification, from which 694,811 particles were selected. After the initial 3D classification, 188,376 particles were selected and subjected to 3D auto-refinement. Next, a soft mask excluding the micelle density was applied and particles were classified into 5 classes without performing alignment. From this, one class with 41,193 particles was selected and further refined, yielding a map at 3.06 Å overall resolution (Supplementary Fig. 5).

For the dataset of mOTOP2 at pH 8.0, 5598 movies were collected, and 5094 movies were selected after motion correction and CTF estimation. 1,230,696 particles were extracted from the selected micrographs and subjected to one round of 2D classification, from which 882,784 particles were selected. After the initial 3D classification, 188,376 particles were selected and subjected to 3D auto-refinement. Next, a soft mask excluding the micelle density was applied and particles were classified into 5 classes without performing alignment. From this, one class with 42,662 particles was selected and further refined, yielding a map at 2.95 Å overall resolution (Supplementary Fig. 6).

For the dataset of mOTOP2 at pH 7.0, 5937 movies were collected, and 5642 movies were selected after motion correction and CTF estimation. 1,501,277 particles were extracted from the selected micrographs and subjected to one round of 2D classification, from which 626,418 particles were selected. After the initial 3D classification, 291,034 particles were selected and subjected to 3D auto-refinement. Next, a soft mask excluding the micelle density was applied and particles were classified into 5 classes without performing alignment. From this, two conformations of the channel for low pH and intermediate states, respectively, were selected for further refinement. In the low pH state, 52,733 particles were then refined and yielded a density map at an overall resolution of 3.22 Å. In the intermediate state, 54,672 particles were then refined and yielded a density map at an overall resolution of 3.79 Å (Supplementary Fig. 7a).

For the dataset of mOTOP2 M374W at pH 8.0, 6145 movies were collected, and 5974 movies were selected after motion correction and CTF estimation. 930,526 particles were extracted from the selected micrographs and subjected to one round of 2D classification, from which 395,899 particles were selected. After the initial 3D classification, 168,336 particles were selected and subjected to 3D auto-refinement. Next, a soft mask excluding the micelle density was applied and particles were classified into 5 classes without performing alignment. From this, one class with 55,837 particles was selected and further refined, yielding a map at 3.12 Å overall resolution (Supplementary Fig. 11).

## Model building, refinement, and validation

The alphafold-predicted structure of CeOTOP8 (Identifier: AF-G5EDX5-F1) or mOTOP2 (Identifier: AF-Q80SX5-F1) was used as the initial model which was then manually adjusted in Coot[32] and refined against the map by using the real space refinement module with secondary structure and non-crystallographic symmetry restraints in the Phenix package[33]. The final model of the CeOTOP8 pH 5.0 structure includes residues 50–87, 94–136, 144–248, 312–345, 397–462, 474–535, and 550–581. The final model of the CeOTOP8 pH 8.0 structure includes residues 47–138, 94–136, 145–270, 303–344, and 358–583. TM7 (residues 366–388) is a well-resolved helix in the high pH CeOTOP8 structure but becomes disordered or mobile in the low pH structure. TM7 is positioned at the outer periphery of the CeOTP8 dimer and does not appear to be important for the channel function. Furthermore, the TM7 helix is well resolved in all other OTOP channel structures including the mOTOP2 structure at low pH. We suspect that low pH destabilizes the packing between TM7 and C-barrel in CeOTOP8, resulting in a mobile TM7 helix. The final model of the mOTOP2 pH 5.0 structure includes residues 28–85, 90–128, 133–198, 231–270, 288–312, 324–354, 364–428, and 487–562. The final model of the mOTOP2 pH 8.0 structure includes residues 27–86, 97–157, 164–195, 238–269, 288–314, 320–354, 361–429, and 487–563. The final model of the pH 7.0 structure from the minor group particles (intermediate state) includes residues 26–53, 59–91, 96–160, 163–196, 238–267, 293–312, 320–355, 366–427, 489–527, and 533–562. The final model of the mOTOP2 M374W pH 8.0 structure includes residues 28–87, 97–157, 164–195, 237–269, 287–314, 320–354, 361–429, and 487–563. The statistics of the geometries of the models were generated using MolProbity[34]. All the figures were prepared in PyMol (Schrödinger, LLC.), UCSF Chimera[35].

## Analysis of mOTOP2 mutant protein expression

The mOTOP2 loss-of-function mutants were cloned into a pEZTBM vector for protein expression using the same construct as the wild-type channel. About 2 μg of the corresponding plasmids were transfected into HEK293F cells grown in a six-well tissue culture dish using Lipofectamine 2000 (Invitrogen). Forty-eight hours after transfection, cells were collected and resuspended in buffer A supplemented with a protease inhibitor cocktail and homogenized by sonication on ice. Protein was extracted with 1% (w/v) DDM supplemented with 0.2% (w/v) CHS by gentle agitation for 2 h. After extraction, the supernatants were collected after a 30 min centrifugation at $20,000 \times g$. Equal volumes of WT and mutant supernatants were loaded on SDS-PAGE followed by western blot. The GFP-mOTOP2 expression level was detected by scanning the SDS-PAGE gel with Bio-Rad ChemiDoc MP system using Alexa Fluor 488 detector. Tubulin was used as internal control for protein expression. For tubulin immunoblotting, the SDS-PAGE gel was transferred onto 0.2-μm nitrocellulose membranes (Pall Life Sciences cat#66485). Membranes were blocked with 5% non-fat milk followed by incubation with anti-tubulin antibody (Cell signaling, cat#86298), 1:2000 dilution. Membranes were then incubated with Goat anti-Mouse IgG (H+L) secondary antibody, Alexa Fluor 680 (Invitrogen, cat#A28183), 1:10,000 dilution, and scanned using Bio-Rad ChemiDoc MP system.

## Electrophysiology

The N-terminal GFP-tagged WT CeOTOP8 and mOTOP2 and their mutants were cloned into a pEGFPC2 vector. About 2 μg of the corresponding plasmids were transfected into HEK293 cells grown in a six-well tissue culture dish using Lipofectamine 2000 (Invitrogen). Forty-eight hours after transfection, cells were dissociated by trypsin treatment and kept in complete serum-containing medium and re-plated on 35 mm tissue culture dishes in a tissue culture incubator until current recording. Patch clamp recordings in the whole-cell configurations was used to measure channel activity. The standard intracellular solution (pipette) contained (in mM): 70 Cesium methanesulfonate (Cs-MS), 2 $MgCl_2$, 1 EGTA, 100 HEPES buffered with Tris, pH 7.4. The extracellular $Na^+$-free solution (bath) contained (in mM): 150 NMDG-MS, 10 HEPES buffered with Tris, pH 7.4. For pH below 6.5, HEPES was replaced with MES buffered with Tris. For pH 8.5, HEPES was replaced with Tris buffered with HCl. The data were acquired using an AxoPatch 200B amplifier (Molecular Devices) and a low-pass analog filter set to 1 kHz. The current signal was sampled at a rate of 20 kHz using a Digidata 1550B digitizer (Molecular Devices) and further analyzed with pClamp 11 software (Molecular Devices). Patch pipettes were pulled from borosilicate glass (Harvard Apparatus) and heat-polished to a resistance of 5–8 MΩ filled with the pipette solution. After the patch pipette attached to the cell membrane, a giga-seal (>10 GΩ) was formed by gentle suction. The whole-cell configuration was formed by short zap or suction to rupture the patch. For continuous current recording, the membrane potential was held at −80 mV. To generate the current and voltage relationship, the membrane potential was held at 0 mV, followed by voltage pulses ramp from −100 to +100 or 120 mV over a 150 ms duration.

## Reporting summary

Further information on research design is available in the Nature Portfolio Reporting Summary linked to this article.

## Data availability

The data that support this study are available from the corresponding author upon request. The cryo-EM density maps of CeOTOP8 and mOTOP2 have been deposited in the Electron Microscopy Data Bank (EMDB) under accession numbers EMD-42214 (CeOTOP8 pH 5.0), EMD-42213 (CeOTOP8 pH 8.0), EMD-42215 (mOTOP2 pH 5.0), EMD-42216 (mOTOP2 pH 8.0), EMD-42217 (mOTOP2 pH 7.0 minor) and EMD-42219 (mOTOP2 M374W pH 8.0). Atomic coordinates have been deposited in the Protein Data Bank (PDB) under accession numbers 8UG5 (CeOTOP8 pH 5.0), 8UG4 (CeOTOP8 pH 8.0), 8UG6 (mOTOP2 pH 5.0), 8UG7 (mOTOP2 pH 8.0), 8UG8 (mOTOP2 pH 7.0 minor) and 8UGA (mOTOP2 M374W pH 8.0). The source data underlying Supplementary Fig. 10a are provided as a Source Data file. Source data are provided with this paper.

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

## Acknowledgements

Single-particle cryo-EM data were collected at the University of Texas Southwestern Medical Center Cryo-EM Facility that is funded by the CPRIT Core Facility Support Award RP170644 and the Howard Hughes Medical Institute Janelia Cryo-EM Facility. N.G. is an HHMI fellow of the Jane Coffin Childs Memorial Fund. This work was supported in part by the Howard Hughes Medical Institute and by grants from the National Institute of Health (R35GM140892 to Y.J.), the Welch Foundation (Grant I-1578 to Y.J.), the National Natural and Science Foundation of China (32071202 and 32271012 to Q.C.), and Tianjin Fund for Distinguished Young Scholars (20JCJQJC00080 to Q.C.).

## Author contributions

N.G. prepared the samples; Y.H. and N.G. performed data acquisition, image processing, and structure determination; W.Z. performed electrophysiology recording; Q.C performed earlier study of CeOTOP8. Y.J supervised the work. All authors participated in research design, data analysis, discussion, and manuscript preparation.

## Competing interests

The authors declare no competing interests.
