## [Peer Review File · Nature Communications]

Structural mechanism of proton conduction in otopetrin
proton channelREVIEWER COMMENTS

Reviewer #1 (Remarks to the Author):

OTOP proton channels are key receptors for sour-tastes, and the precise mechanism of proton gating remains elusive. Previously, Kei et al. reported nanodisc structures of zebrafish OTOP1 and chicken OTOP3 at pH8 (PMID: 31160780), along with all-atom molecular dynamics simulations leading to the identification of three putative pores for proton conduction. Around the same time, the manuscript's authors' group also published the OTOP3 structure from clawed frog at pH 8 (PMID: 30973323), with electrophysiological and mutagenic analyses. Currently, key questions regarding OTOPs include structural evidence for proton activation and/or desensitization, the identification of proton conduction pathway. In this manuscript, the authors describe the first work of two OTOPs at pH 8 and pH 5, in GDN detergent, and analyzed the proton interplay through electrophysiology and mutagenesis. While this study produced great cryo-EM structures, there are several concerns which need to be addressed.

Major concerns:

In this study the non-functional channel constructs are used for structural studies. Therefore, any conformational changes induced by acidic pH would not represent the true activation and proton conduction mechanism. Therefore, "...self-inhibition allows us to capture the WT CeOTOP8 channel structure in a transiently proton-transferring..." this statement does not make sense. The pH sensors are not fully mapped in the channel. The present setup is not physiologically relevant where both sides of channel are influenced by pH changes simultaneously. The residues present at the cytoplasmic side might get influenced by the low pH (at pH 5). Therefore, the conformational changes that is observed could be just due the change of pH effect nothing to do with activation mechanism. The pKa of Glutamate side chain is about 4.25. Therefore, in both pH 5 and 8, the side chain will be deprotonated. Therefore, it not clear how it is carrying proton which is then transferred to histidine.

CeOTOP8 and mOTOP2 have been used to depict the working mechanism of activation and proton transfer. However, the proposed mechanism could be totally flawed if the pH sensitivity of both channels at the cytoplasmic site differs. The conformational changes can be truly compared if the cytoplasmic side is not exposed to pH changes.

The proposed mechanism is also not convincing according to the provided data. According to the discussions for CeOTOP8 pH 8 and pH 5 models (line 15 page 6: "Thus, this structural transition appears..."), the retreat of F563 from the space for E325/H567 communication is due to the helix-to-loop transformation of N-terminal part of TM12, then apparently it should be the first incident for TM12 to uncoil and F563 to swing away upon proton transport. In other words, the N-term of TM12 should remain a loop in the presence of low external pH to prevent F563 from occupying the space instead of coiling/uncoiling in a circle (figure 7). This needs to be clarified.

After checking the densities of few residues, it seems like the data (at pH8 especially CeOTOP8) is the mixture of two states. The author should try to separate this by further classifications.

The present study also demonstrated the structure of loss of function mutant M374W at pH8.

It will be interesting to solve the structure at pH 5 to investigate conformational changes induced by low pH.

According to this study the accumulation of the local protons stalls the proton conduction. Does that mean the structures captured here at low pH are desensitized conformations?

Minor concerns:

“demonstrates” instead of “demonstrate” in the second last sentence of the abstract.

The “proceeding” on the second last line on page 4 should be “preceding” instead. Likewise, the “proceeding” on the fifth line of page 7.

Map densities for E325 can be presented in figure 3a for both pH conditions to support the story.

The rugged density mesh in figure 3c can be replaced by proper transparent densities for aesthetic purpose.

Densities for the lipids at the pore have to be modelled.

In method “..with a dose rate of $1e^{-}/\text{\AA}^2/\text{frame}$ for a total dose of $60e^{-}/\text{\AA}^2$..” \AA^2 will be $\text{\AA}(\text{square})^2$.

“on a GIF-Quantum energy filter. Data were collected using Falcon 4 camera with a resolution pixel size of..” resolution should be super resolution.

Reviewer #2 (Remarks to the Author):

This study describes the structures of two otopetrin channels that constitutively conduct protons at various extracellular pH, *C. elegans* OTOP8 and mouse OTOP2. Each structure was determined by Cryo-EM at pH 5 and pH 8. The comparison between the structures suggests a swing motion of the C-barrel hinged at the external ends of TM9 and TM10. It also shows different positions of conserved glutamate in TM6 and histidine in TM12 that could mediate the transfer of protons across the membrane. Based on this comparison, the authors propose a conduction mechanism in which protons move at the interface between the N- and C-barrel as the protein transitions through multiple states. I do not have concerns about the experimental data, but the following points regarding data interpretation need to be addressed:

The authors assume that because CeOTOP8 and mOTOP2 are both active at different pH, the structures captured by Cry-EM are from conductive states. While it is reasonable to expect states other than resting/closed, it is unclear how conductive states are distinguished from desensitized/inactive ones.

It is interesting that none of the determined structures shows an open transmembrane

conduction pathway, as one would expect from an open channel. The presence of a constriction in the pathway would also be consistent with an open channel, as long as protons can move across the constriction. In the structures the pathway appears to be sterically blocked at different locations. But, the presence of water molecules moving inside the channel could reduce the gap between intracellular and extracellular cavities allowing protons to reach titratable residues deep inside the protein. Were all-atom MD simulations in water and membrane environment carried out to see where water molecules may go?

The authors propose a proton conduction mechanism that involves consecutive conformational changes (Fig. 7). The OTOP protein binds an extracellular proton in an outward-facing conformation (state 1), then transitions to consecutive occluded conformations in which the proton has no access to either side of the membrane (states 2 – 4), until another transition converts the protein into an inward-facing conformation (state 5) from which the proton is released (state 6). This mechanism is consistent with how a transporter functions. However, the authors refer to OTOPs as proton channels and show data consistent with channel function.

Is there experimental evidence that OTOPs are in fact transporters? Would the proposed mechanism imply that proton flux is rate limited by how fast the protein can move through the conformational changes in the transport cycle? Is there any evidence of current saturation with proton concentration that can support this prediction? These questions should be addressed in the discussion section.

Intra-barrel conduction pathways in OTOP proteins have been previously proposed. The authors state that their finding “negates our initial hypothesis suggesting that each barrel forms an independent proton passage.” Since the study only focuses on a transport mechanism involving the inter-barrel pathway, the authors should discuss how the structures of CeOTOP8 and mOTOP2 look like inside each barrel and what evidence support excluding these proton regions from mediating proton transport.

Response to Reviewer #1:

We appreciate reviewer #1's insightful comments and helpful suggestions. The points raised by the reviewer are addressed as follows:

Reviewer #1 (Remarks to the Author):

OTOP proton channels are key receptors for sour-tastes, and the precise mechanism of proton gating remains elusive. Previously, Kei et al. reported nanodisc structures of zebrafish OTOP1 and chicken OTOP3 at pH8 (PMID: 31160780), along with all-atom molecular dynamics simulations leading to the identification of three putative pores for proton conduction. Around the same time, the manuscript's authors' group also published the OTOP3 structure from clawed frog at pH 8 (PMID: 30973323), with electrophysiological and mutagenic analyses. Currently, key questions regarding OTOPs include structural evidence for proton activation and/or desensitization, the identification of proton conduction pathway. In this manuscript, the authors describe the first work of two OTOPs at pH 8 and pH 5, in GDN detergent, and analyzed the proton interplay through electrophysiology and mutagenesis. While this study produced great cryo-EM structures, there are several concerns which need to be addressed.

Major concerns:

In this study the non-functional channel constructs are used for structural studies. Therefore, any conformational changes induced by acidic pH would not represent the true activation and proton conduction mechanism. Therefore, "...self-inhibition allows us to capture the WT CeOTOP8 channel structure in a transiently proton-transferring..." this statement does not make sense. The pH sensors are not fully mapped in the channel. The present setup is not physiologically relevant where both sides of channel are influenced by pH changes simultaneously. The residues present at the cytoplasmic side might get influenced by the low pH (at pH 5). Therefore, the conformational changes that is observed could be just due the change of pH effect nothing to do with activation mechanism. The pKa of Glutamate side chain is about 4.25. Therefore, in both pH 5 and 8, the side chain will be deprotonated. Therefore, it not clear how it is carrying proton which is then transferred to histidine.

A key finding of our study is that different from conventional channels, OTOP channels do not have a particular conformation that represents an active or conductive state. Instead, the proton conduction of OTOP channels involves multiple steps. The conformational changes observed in our CeOTOP8 or mOTOP2 structures only represent part of the process rather than the

complete process. The inhibition of CeOTOP8 by its N-terminal inter-subunit interaction stabilizes the channel in one particular step where extracellular Glu325 can directly interact with intracellular His567. Based on all the OTOP structures from this and previous studies, it makes sense that this bridging interaction represents a transient state where proton transfer can occur between Glu and His.

We don't know the pH sensor in CeOTOP8 or mOTOP2. We suspect the conformational change is driven by the protonation state of the conserved Glu (i.e. Glu325 in CeOTOP8). It is worth noting that different from OTOP1 and OTOP3 which require acidic extracellular pH to be active, both CeOTOP8 and mOTOP2 are active even at higher extracellular pH (i.e. pH 7.4) as demonstrated in our electrophysiology experiments. How OTOP1 and OTOP3 sense extracellular pH is still an open question.

For simplicity, we only present the representative structures at pH 5 and 8 in this study. We have actually determined the structures of both channels at various pHs (5.0, 6.0, 7.0, 7.4, 8.0, and 9.0) and their structural behaviors are remarkably stable. Other than the observed structural changes at the extracellular part of TM6 or TM12 as discussed in the paper, the cytosolic part of the channel remains virtually identical at different pHs. Although we cannot rule out the possibility that the intracellular pH difference influences the conformational change on the extracellular side, we have not seen obvious evidence to support this possibility based on all the available OTOP structures.

The pKa of acidic residues can vary significantly depending on their microenvironment. The pKa of an acidic residue buried in a hydrophobic environment increases significantly as compared to that exposed in the solvent. For instance, in bacteriorhodopsin, a buried Glu critical for proton pumping has a high pKa value between 9 and 12 (PMID: 8672439, 21709261, 33817510). Additionally, a pKa of approximately 8.7 has been measured for a buried Asp residue in a model transmembrane helix (PMID: 15236588). We have included this point in the revision.

CeOTOP8 and mOTOP2 have been used to depict the working mechanism of activation and proton transfer. However, the proposed mechanism could be totally flawed if the pH sensitivity of both channels at the cytoplasmic site differs. The conformational changes can be truly compared if the cytoplasmic side is not exposed to pH changes.

We are not exactly sure what the pH sensitivity at the cytoplasmic site refers to. As in our response to the previous point, we have determined the structures of both channels at various pHs from 5 to 9. Combined with previous studies, we have not seen evidence to suggest that the intracellular pH difference influences the conformational change observed on the extracellular side. From structural and functional studies of different OTOPs, the main influence of intracellular pH appears to be the protonation of the conserved Histidine (H567 in CeOTOP8 and H551 in mOTOP2) which in turn determines if this histidine can be a proton acceptor or not. Also, from our

electrophysiological analysis, the functional properties of CeOTOP8 and mOTOP2 are very similar.

The proposed mechanism is also not convincing according to the provided data. According to the discussions for CeOTOP8 pH 8 and pH 5 models (line 15 page 6: “Thus, this structural transition appears...”), the retreat of F563 from the space for E325/H567 communication is due to the helix-to-loop transformation of N-terminal part of TM12, then apparently it should be the first incident for TM12 to uncoil and F563 to swing away upon proton transport. In other words, the N-term of TM12 should remain a loop in the presence of low external pH to prevent F563 from occupying the space instead of coiling/uncoiling in a circle (figure 7). This needs to be clarified.

As discussed in an earlier response, the proton transfer in OTOP is a multi-step process. The helix-to-loop structural transition on TM12 only represents one step that appears to be prerequisite for the formation of the proton-transferring bridge between Glu325 and His567. If the N-terminal part of TM12 remains a loop structure, F563 will prevent Glu325 from flipping back to the extracellular side. Therefore, upon proton transfer from Glu325 to His567, it is necessary for N-terminus of TM12 to reform the helix so that Glu325 can flip back and get re-protonated from extracellular side. Otherwise, the proton conduction would be stalled if TM12 remains coiled. One reason why the Glu-His bridge does not form at lower pH is because His567 is protonated and cannot serve as an acceptor (analogous to low pH desensitization). We have modified the discussion to clarify.

After checking the densities of few residues, it seems like the data (at pH8 especially CeOTOP8) is the mixture of two states. The author should try to separate this by further classifications.

We thank the reviewer for this suggestion. Indeed, in the pH 8 structure of CeOTOP8, the density around H567 indicates two possible conformations. However, the difference is subtle and in a small local area. We were not able to separate them through further classifications.

The present study also demonstrated the structure of loss of function mutant M374W at pH8. It will be interesting to solve the structure at pH 5 to investigate conformational changes induced by low pH.

M374 is located at the dimer interface of mOTOP2 far from the area where conformational changes occur and its mutation should not affect the observed conformational change on TM6. The M374W mutant structure is identical to the wild type at pH 8. We determined its structure, mainly to demonstrate that its loss of function is not caused by any structural changes. As the

mOTOP2 structures at the dimer interface are identical at low and high pH, we expect the M374W mutant structure will be the same as WT at pH 5.

According to this study the accumulation of the local protons stalls the proton conduction. Does that mean the structures captured here at low pH are desensitized conformations?

Indeed, as we discussed in the paper, the low pH structure likely represents a desensitized state. It is worth pointing out that the desensitization is caused by the protonation of the intracellular histidine, preventing it from accepting the proton from Glu. This histidine-mediated desensitization does not appear to affect the extracellular conformational changes.

Minor concerns:

“demonstrates” instead of “demonstrate” in the second last sentence of the abstract.

It has been corrected.

The “proceeding” on the second last line on page 4 should be “preceding” instead. Likewise, the “proceeding” on the fifth line of page 7.

Corrected as suggested.

Map densities for E325 can be presented in figure 3a for both pH conditions to support the story.

The densities for E325 have been included in Figure 3a as suggested.

The rugged density mesh in figure 3c can be replaced by proper transparent densities for aesthetic purpose.

We have changed the representation of the density as suggested.

Densities for the lipids at the pore have to be modelled.

The lipid densities at the central hole of the channel dimer are very heterogeneous in length and size, making it impossible to define their identities properly. We therefore decided not to model these lipids.

In method “..with a dose rate of $1e^{-}/\text{\AA}^2/\text{frame}$ for a total dose of $60e^{-}/\text{\AA}^2$..” \AA^2 will be $\text{\AA}(\text{square})^2$.

Corrected as suggested.

“on a GIF-Quantum energy filter. Data were collected using Falcon 4 camera with a resolution pixel size of..” resolution should be super resolution.

We have made the correction.

Response to Reviewer #2:

We appreciate reviewer #2's positive comments and constructive suggestions. The points raised by the reviewer are addressed as follows:

Reviewer #2 (Remarks to the Author):

This study describes the structures of two otopetrin channels that constitutively conduct protons at various extracellular pH, *C. elegans* OTOP8 and mouse OTOP2. Each structure was determined by Cryo-EM at pH 5 and pH 8. The comparison between the structures suggests a swing motion of the C-barrel hinged at the external ends of TM9 and TM10. It also shows different positions of conserved glutamate in TM6 and histidine in TM12 that could mediate the transfer of protons across the membrane. Based on this comparison, the authors propose a conduction mechanism in which protons move at the interface between the N- and C-barrel as the protein transitions through multiple states. I do not have concerns about the experimental data, but the following points regarding data interpretation need to be addressed:

The authors assume that because CeOTOP8 and mOTOP2 are both active at different pH, the structures captured by Cry-EM are from conductive states. While it is reasonable to expect states other than resting/closed, it is unclear how conductive states are distinguished from desensitized/inactive ones.

A key finding of our study is that different from conventional channels, OTOP channels do not have a particular conformation that represents an active or conductive state. Instead, the proton conduction process of OTOP channels involves multiple steps. The conformational changes observed in our CeOTOP8 or mOTOP2 structures only represent part of the process rather than the complete process. The channel can be inactive if it is trapped in one particular step. The channel is desensitized at lower intracellular environment because of the protonation of the conserved His, preventing it from accepting the proton from Glu. This histidine-mediated desensitization does not appear to affect the extracellular conformational changes. We have clarified these points in the discussion.

It is interesting that none of the determined structures shows an open transmembrane conduction pathway, as one would expect from an open channel. The presence of a constriction in the pathway would also be consistent with an open channel, as long as protons can move across the constriction. In the structures the pathway appears to be sterically blocked at different locations. But, the presence of water molecules moving inside the channel could

reduce the gap between intracellular and extracellular cavities allowing protons to reach titratable residues deep inside the protein. Were all-atom MD simulations in water and membrane environment carried out to see where water molecules may go?

In a previous study, all-atom molecular dynamics simulations were performed on zebrafish Otop1 and chicken Otop3 in a mixed lipid bilayer to examine water penetration and explore potential proton permeation pathways. The results point to three possible pathways: the aqueous vestibules in the N and C domains, and the inter-domain interface between N and C domains. The water wires for both N and C domains have clear break with hydrophobic constriction. The water wire at the inter-domain interface appears to be more continuous but also has constrictions formed by hydrophobic residues. Our results confirm that the inter-domain interface forms the proton conduction pathway and the movement of the conserved Glu and His mediate the proton permeation. As discussed in our previous response, unlike conventional channels, our study suggests that OTOP channel conduction is a multi-step process and the channel does not have a specific conformation representing an open state.

The authors propose a proton conduction mechanism that involves consecutive conformational changes (Fig. 7). The OTOP protein binds an extracellular proton in an outward-facing conformation (state 1), then transitions to consecutive occluded conformations in which the proton has no access to either side of the membrane (states 2 – 4), until another transition converts the protein into an inward-facing conformation (state 5) from which the proton is released (state 6). This mechanism is consistent with how a transporter functions. However, the authors refer to OTOPs as proton channels and show data consistent with channel function.

We agree with the reviewer that the way the proton permeates through the OTOP channel is somewhat like a transporter. In light of its functional behavior: fast passive permeation of proton down the electrochemical gradient without counter ions or energy (ATP), we still should classify it as a channel.

Is there experimental evidence that OTOPs are in fact transporters? Would the proposed mechanism imply that proton flux is rate limited by how fast the protein can move through the conformational changes in the transport cycle? Is there any evidence of current saturation with proton concentration that can support this prediction? These questions should be addressed in the discussion section.

As mentioned in our last response, we would classify OTOPs as channels because of their functional behavior despite their transporter-like proton conduction mechanism. We agree with the reviewer that the rate of conformational changes in the cycle would determine the proton flux rate. Indeed, that is what we observed in the electrophysiology of both CeOTOP8 and mOTOP2. As shown in Fig1a and 4b, the proton currents of both channels become saturated once the extracellular pH reaches about 5.5. We have revised the discussion to incorporate these points.

Intra-barrel conduction pathways in OTOP proteins have been previously proposed. The authors state that their finding “negates our initial hypothesis suggesting that each barrel forms an independent proton passage.” Since the study only focuses on a transport mechanism involving the inter-barrel pathway, the authors should discuss how the structures of CeOTOP8 and mOTOP2 look like inside each barrel and what evidence support excluding these proton regions from mediating proton transport.

The conformational changes observed in this study only affect the inter-barrel interface. We observe no structural difference within each barrel among all the available OTOP structures (from both this study and previous ones). Both barrels are constricted at the intracellular half of the channel in all the structures. In our previous study, we also did extensive mutations within each barrel and most of the mutant channels work fine. On the contrary, mutations at the conserved Glu or His at the inter-barrel interface result in a complete loss of function. Since our statement of “negates our initial hypothesis suggesting that each barrel forms an independent proton passage.” is a bit out of text in the manuscript, we decided to remove it in the revision.

REVIEWER COMMENTS

Reviewer #1 (Remarks to the Author):

The authors' response does justify that these provided OTOP structures represent different functional status despite the extra/intracellular pH environments being not native-like. Here are some concerns listed below which need to be addressed.

Line 123: "... the functional construct was unexpectedly difficult to over-express and purify...". If the construct refers to the $\Delta 1$ -57CeOTOP8 which shows robust currents at near neutral pH, then it is not very surprising that it does not express well since the stronger proton flow might kill cells. Interestingly, whole-cell recordings of mOTOP2 and $\Delta 1$ -57CeOTOP8 seem to have similar patterns, with characteristic currents from pH6 to 4.5 (Fig. 1a and Fig. 4b), while mOTOP2 could be expressed and purified, $\Delta 1$ -57CeOTOP8 could not be expressed. Is it because $\Delta 1$ -57CeOTOP8 is active even at symmetrical pH7.4 and mOTOP2 is not? If so, the $\Delta 1$ -57 seems to be too active to be called "functional" but rather unnecessarily leaky.

Line 181: "... this histidine also plays a central role in OTOP proton conduction and its mutations to Ala or Gln led to the loss of function...". According to the reference (18), I suggest the use of "complete loss of function" or "abolished function" instead to reduce ambiguity, as the extent of "loss of function" can be varied and only complete loss can manifest the residue as an indispensable part of proton-transfer path.

In Figure 5c, pH8.0, the distances of H-bonding and salt bridge can be labelled near the dashed lines if possible.

One important hypothesis made by the authors is that the C-barrel of mOTOP2 swings during conducting to let Glu250 and His551 approach each other to allow proton transfer. However, the rationale comes only from the comparison between CeOTOP and mOTOP, which have overall structural differences, and that certain W mutants at barrel interface abolished channel function. They cannot be considered as strong direct proof in my opinion. The hypothesis that C-barrel swinging may be true for OTOP, but it cannot be supported by the structures reported here and may be verified only if it is observed for one protein, either within one dataset or under different conditions.

According to Figures S1, S2, S5 and S6, the maps improvements from 3D classification without alignment to the final map after CTF refinement and polish are very drastic, while the difference between 3D classification classes are not that much. So authors should try to use all the pre-3D classification particles to align, CTF refine, and polish and then do 3D classifications without alignment to see whether more useful particles and different conformations can emerge? How about performing focused classification for subunits? I wonder whether there are different conformation states in each dataset especially the pH5 ones. If swinging and non-swinging conformations co-exist in one dataset, that will be convincing enough for the hypothesis that one barrel swings against the other.

Figure S9b, arrows are not shown for the peaks of protein although written in the legend. I suppose all arrows should refer to the ~15mL peak and thus justify the mutants all form dimers like the WT.

Line 388: "For sample at pH 5.0, approximately 0.06% (v/v) glacial acetic acid was added directly to the sample right before grid preparation." If I calculated correctly, 0.06% glacial

acetic acid corresponds to ~10mM, which does not seem to be able to reduce pH to 5.0 in a 20mM Tris pH8 buffer. Can you give some justifications for this procedure? Similarly in the authors' response, "... determined the structures of both channels at various pHs (5.0, 6.0, 7.0, 7.4, 8.0 and 9.0) and their structural behaviors are remarkably stable", I wonder how these pH conditions are applied and the reliability of it. In case the particle numbers are low for each dataset, you may consider combining some/all different-pH datasets together then classify to search for novel conformations.

Figure S10, the "FSC=0.143" dashed line seems to be > 0.15.

Reviewer #2 (Remarks to the Author):

My previous comments have been addressed appropriately. I have no further concerns.

Response to Reviewer #1:

We appreciate reviewer #1's insightful comments and helpful suggestions. The points raised by the reviewer are addressed as follows:

Reviewer #1 (Remarks to the Author):

The authors' response does justify that these provided OTOP structures represent different functional status despite the extra/intracellular pH environments being not native-like. Here are some concerns listed below which need to be addressed.

Line 123: "... the functional construct was unexpectedly difficult to over-express and purify...". If the construct refers to the $\Delta 1-57\text{CeOTOP8}$ which shows robust currents at near neutral pH, then it is not very surprising that it does not express well since the stronger proton flow might kill cells. Interestingly, whole-cell recordings of mOTOP2 and $\Delta 1-57\text{CeOTOP8}$ seem to have similar patterns, with characteristic currents from pH6 to 4.5 (Fig. 1a and Fig. 4b), while mOTOP2 could be expressed and purified, $\Delta 1-57\text{CeOTOP8}$ could not be expressed. Is it because $\Delta 1-57\text{CeOTOP8}$ is active even at symmetrical pH7.4 and mOTOP2 is not? If so, the $\Delta 1-57$ seems to be too active to be called "functional" but rather unnecessarily leaky.

Actually, both mOTOP2 and $\Delta 1-57\text{CeOTOP8}$ are active at symmetrical pH 7.4 (Fig 1b and Fig. 4a). We don't know the exact cause for the difficulty in overexpressing the $\Delta 1-57\text{CeOTOP8}$ construct. We notice that under the same recording condition, the negative membrane potential elicits a much larger inward proton current in the $\Delta 1-57\text{CeOTOP8}$ -expressing HEK cell than in the mOTOP2-expressing HEK cell. We suspect the higher activity of $\Delta 1-57\text{CeOTOP8}$ could potentially cause more damage to the cells and contribute to the over-expression problem.

Line 181: "... this histidine also plays a central role in OTOP proton conduction and its mutations to Ala or Gln led to the loss of function...". According to the reference (18), I suggest the use of "complete loss of function" or "abolished function" instead to reduce ambiguity, as the extent of "loss of function" can be varied and only complete loss can manifest the residue as an indispensable part of proton-transfer path.

We made the change as suggested.

In Figure 5c, pH8.0, the distances of H-bonding and salt bridge can be labelled near the dashed lines if possible.

Distance labels added

One important hypothesis made by the authors is that the C-barrel of mOTOP2 swings during conducting to let Glu250 and His551 approach each other to allow proton transfer. However, the rationale comes only from the comparison between CeOTOP and mOTOP, which have overall structural differences, and that certain W mutants at barrel interface abolished channel function. They cannot be considered as strong direct proof in my opinion. The hypothesis that C-barrel swinging may be true for OTOP, but it cannot be supported by the structures reported here and may be verified only if it is observed for one protein, either within one dataset or under different conditions.

Although unable to capture the two proposed C-barrel conformations within the same protein, we did observe an intermediate conformation in one of the mOTOP2 data from pH 7 sample that was not discussed in the earlier version. The mOTOP2 structures were determined at various pH conditions (5.0, 7.0, 8.0, and 9.0). Structures at pH 8 and 9 are virtually identical. At neutral pH, however, the mOTOP2 protein becomes more dynamic and its particles can be classified into two groups. The structure from the major group of particles (at 3.22 Å) is similar to that determined at pH 5.0. The mOTOP2 structure obtained from the minor group of particles (3.79 Å) adopts a conformation different from those obtained at low and high pH. Its C-barrel moves to an intermediate state between the two main C-barrel conformations observed in mOTOP2 and CeOTOP8, suggesting a dynamic motion of the C-barrel in mOTOP2. We have included this information in the second revision of the manuscript.

According to Figures S1, S2, S5 and S6, the maps improvements from 3D classification without alignment to the final map after CTF refinement and polish are very drastic, while the difference between 3D classification classes are not that much. So authors should try to use all the pre-3D classification particles to align, CTF refine, and polish and then do 3D classifications without alignment to see whether more useful particles and different conformations can emerge? How about performing focused classification for subunits? I wonder whether there are different conformation states in each dataset especially the pH5 ones. If swinging and non-swinging conformations co-exist in one dataset, that will be convincing enough for the hypothesis that one barrel swings against the other.

We have tried multiple data-processing strategies, but none of them allow us to classify the particles into different conformations for most of the data. We have also tried the suggested strategy of using all the pre-3D classification particles to align, CTF refine, and polish, followed by 3D classifications on two mOTOP2 datasets (pH 5 and 8), and got the same results. As discussed in our response to the previous point, one exception is the mOTOP2 data collected at pH 7.0. We were able to classify the particles from this dataset into two classes. One class yielded a structure of mOTOP2 with its C-barrel in an intermediate state between the two main C-barrel conformations observed in mOTOP2 and CeOTOP8. This information has been included in the second revision of the manuscript.

Figure S9b, arrows are not shown for the peaks of protein although written in the legend. I suppose all arrows should refer to the ~15mL peak and thus justify the mutants all form dimers like the WT.

We have marked the peaks with arrows in the revised figure.

Line 388: “For sample at pH 5.0, approximately 0.06% (v/v) glacial acetic acid was added directly to the sample right before grid preparation.” If I calculated correctly, 0.06% glacial acetic acid corresponds to ~10mM, which does not seem to be able to reduce pH to 5.0 in a 20mM Tris pH8 buffer. Can you give some justifications for this procedure? Similarly in the authors’ response, “... determined the structures of both channels at various pHs (5.0, 6.0, 7.0, 7.4, 8.0 and 9.0) and their structural behaviors are remarkably stable”, I wonder how these pH conditions are applied and the reliability of it. In case the particle numbers are low for each dataset, you may consider combining some/all different-pH datasets together then classify to search for novel conformations.

For protein samples at pHs 7.0, 7.4, 8.0, and 9.0, a 20 mM Tris-HCl buffer at the corresponding pH was used throughout the entire protein purification and sample preparation process. For protein sample at lower pH (i.e. pH 5.0), an appropriate volume of acetic acid stock (1%) was added to the purified protein. To determine the amount of acetic acid needed to adjust the pH of the protein sample, we performed a titration using a larger volume of the buffer solution (with 20 mM Tris buffer at pH 8) used for protein purification. When titrated with 50mL of buffer solution, 30 μ L of pure glacial acetic acid is needed to achieve pH 5, equivalent to a 0.06% (v/v) final concentration. Thus, the protein sample at pH 5.0 was obtained by adding acetic acid to a 0.06% (v/v) final concentration to the purified protein before grid preparation.

Figure S10, the “FSC=0.143” dashed line seems to be > 0.15.

Corrected in the revised figure.